# The Classification of Altruism Forms South Korean Coaches’ Perspective

**DOI:** 10.3390/bs14090802

**Published:** 2024-09-11

**Authors:** Namki Lee, Yucheon Kim

**Affiliations:** Department of Counseling and Coaching, Dongguk University, 30, Pildong-ro 1 gil, Jung-gu, Seoul 04620, Republic of Korea; nearman7@dgu.ac.kr

**Keywords:** altruism, coach, Q methodology, subjectivity

## Abstract

Altruism is an important element that enables coaches to achieve their clients’ coaching goals. Using Q methodology, which enables the examination of individuals’ subjectivity, this study investigated South Korean coaches’ perceptions of altruism. Through a literature review, interviews, and surveys, 204 statements were collected from the Q population, and 40 statements were selected to form Q Samples. P samples were organized with 31 coaches registered with the Korea Coach Association, and principal component factor analysis using the Quanl program was applied to assess the data. Based on the results of this study, South Korean coaches’ perceptions of altruism were sorted into four types: type 1, ‘a perception type that considers altruism to lead to respect for the client’s presence’; type 2, ‘a perception type that considers altruism to be meaningful when practiced’; type 3, ‘a perception type that considers altruism is a mindset that leads to the development of happiness’; and type 4, ‘a perception type that considers altruism to be a basic attitude that coaches should have’. The results of this study will further clarify coaches’ perceptions of altruism, enabling them to receive the necessary training, develop their character, and achieve inner maturity. This, in turn, will help them improve their attitudes towards clients and further enhance their ethics and professionalism.

## 1. Introduction

When we understand another person’s situation, feelings of altruism that inspire us to help them are developed, also making us happy as a result [1]. This is because emotions based on displays of humanity, such as altruism, make us look at others positively and feel happy accordingly [2]. Thus, wise people demonstrate emotional maturity, including altruism, which fosters a deep concern for other human beings [3]. In addition, positive feelings promote health by pushing out negative feelings [4], and altruism also reduces anxiety and fear, thereby making flight responses disappear and leading to health and happiness [5].

Altruism can also be defined as acting to benefit others [6]. Furthermore, altruism is sometimes explained as caring for others without expecting personal gain [7]. The emotional aspect of altruism is sometimes explained as a state in which a helping mindset and empathy are combined [8], as well as a motivational state with the goal of improving the happiness of others [9]. Thus, the term ‘altruism’ is commonly heard and used in everyday life; however, it can be defined in various ways depending on the perspective [10].

Altruism, which refers to thinking or acting to benefit others, can be an important element in situations where relationships with clients are formed, such as counseling or coaching [11]. Researchers have provided different definitions of the concept of coaching, and it varies somewhat depending on the field [12]. However, in general, coaching is the act of a coach helping a client find ways to become more effective by themselves [12,13]. Therefore, coaching reflects skills used to guide an individual or group to go from where they currently are to a desired, more satisfactory point [14]. In areas where clients’ development must be promoted, such as coaching, positive motivation to help others, such as altruism, is important [15].

Of course, coaches receive financial compensation from their clients for their own benefit. However, altruistic behavior is defined as behavior that is motivated by concern for the well-being of the recipient of the beneficial action rather than the well-being of the person engaging in the behavior [16]; therefore, as long as concern for others is the primary motivation, an act of altruism can be both ethical and truly altruistic, even if it is motivated by the monetary interest of the person engaging in altruism, including the motivation for compensation [17].

Based on Rogers’ [18] concept of the person-centered approach, counseling emphasizes the importance of empathy as a core qualification that counselors must have to form therapeutic relationships with clients, and coaching has also accepted the counseling theory as such to regard empathy as a basic qualification of coaches [19]. However, although feeling empathy can trigger altruistic motivation to help others [20], and people with high levels of empathy tend to be more altruistic [21], empathy is a close emotional understanding of others, whereas altruism is performing ‘active and practical actions’ for others that go beyond ‘emotional experiences’, such as empathy [22].

Feehily [23] stated that in coaches’ growth processes, a transition in coaching motivation occurs from the initial self-centered motivation to altruistic motivation for the benefit and welfare of clients. Thus, coaches must ultimately become concerned with clients’ interests and welfare. In addition, if altruism is understood as an element that leads to coaches having better relationships with their clients, then coaches can use altruism as a means to promote positive changes in those clients [24].

Although coaches can use their inherent or learned altruism to understand clients and foster relationships with them [22], research to provide theoretical support for this concept is insufficient [22,25]. Furthermore, few studies have addressed coaches’ altruistic motivation [26], while no studies to date have addressed coaches’ perceptions of altruism, which is the aim of this study.

Providing a concrete definition of altruism is difficult as every individual has different subjective perceptions of it. To address this issue, Q methodology enables individual subjectivity to be explored, which refers to a person’s thoughts and perceptions about a certain topic [27]. Q methodology is an appropriate study method for the present work, as it is one of the most well-developed paradigms for investigating human subjectivity [28]. It was first introduced by Stephenson [29] in 1935 and is a mixed quantitative and qualitative methodology designed to better understand individuals’ subjective views regarding a phenomenon [30,31]. When Q methodology is used, the possibility of expressing the subjectivity perceived by study participants can be increased [32], and equal opportunities for study participants to express their thoughts can be provided [31]. Thus, even minority voices can be well explored without omission [33,34].

This study aims to examine how altruism is perceived by South Korean coaches. Specifically, this study applies Q methodology, a method that enables the user to thoroughly investigate the subjective perceptions of various individuals. In addition, this study analyzes the types of altruism perceived by South Korean coaches and is intended to help South Korean coaches improve their relationships with clients and their professional capabilities.

In this study, participants were randomly recruited from an online community of coaches who are members of the Korea Coach Association to explore their perceptions of altruism. The random selection of participants was intended to increase the likelihood that perceptions of altruism would be expressed in various ways, without being generalized based on the participants’ specific conditions [31,33].

Research question 1: What are the types of perceptions South Korean coaches have of altruism?

Research question 2: What are the characteristics of South Korean coaches’ perceptions of altruism?

## 2. Study Method

The overall process through which this study was conducted is summarized in Table 1.

### 2.1. Organization of Q Population

A Q population refers to the concourse of individuals’ subjective perceptions collected for a Q study. To create a Q population, methods such as related literature reviews, interviews, and questionnaire surveys are used, which were the methods used in the present study. Regarding the related literature, major papers on the topic of altruism were searched to extract 96 definitions and expressions of altruism. Semi-structured interviews were conducted on the topic of altruism with 5 coaches, and 38 statements were extracted. The questionnaire survey surveyed 13 coaches about their perceptions of altruism using structured questions to extract 70 statements. As a result, a Q population of 204 statements in total was formed.

For reference, the coach certifications in Table 2 were divided into certifications issued by the Korea Coach Association, which issues certifications divided into level 1 KAC (Korea Associate Coach), level 2 KPC (Korea Professional Coach), and level 3 KSC (Korea Supervisor Coach).

The Korea Coach Association (KCA) is a professional organization for coaches across various professions in Korea, aiming to support the development and professionalization of the coaching field. While some KCA members work exclusively as professional coaches, many engage in additional coaching activities within their professions to further enhance their expertise and depth. The Korea Coach Association offers different levels of certification based on the number of hours completed in educational programs and coaching practice. To obtain a certification, you must complete a coaching program provided by an educational institution accredited by the Korea Coach Association and fulfill a required number of coaching practice hours. Upon completing the course and passing both written and practical exams, you will be granted a coaching certification at the respective level.

### 2.2. Selection of Q Samples

Q samples are statements extracted from the Q population. Robinson and Curry [35] proposed the Altruism Development Model (ADM), which divides altruism into four factor categories: biological, cognitive, social, and religious/spiritual factors. Based on their sorting criteria for altruism, the 204 statements from the Q population were divided based on the four factors. The Q population statements were repeatedly analyzed to remove duplicate items, and secondary samples were selected under the guidance of an expert who lectures on Q methodology at a graduate school. Next, four doctoral students who took lectures on Q methodology reviewed it. Sentences with ambiguous meanings or that were otherwise difficult to understand were revised through collaborative discussions. Through pilot tests of the Q samples mentioned above, 40 final Q statements were selected, as shown in Table 3.

### 2.3. Composition of P Samples

A P sample is a person who sorts the extracted Q statements based on his or her subjective perception. The P samples in this study consisted of 31 coaches registered with the Korea Coaches Association. Before conducting the survey, the purpose and procedures of the study were fully explained to the P samples, and their informed consent was obtained.

### 2.4. Q Sorting and Data Analysis

Q sorting refers to when P samples sort Q samples. Before the Q sorting began, four doctoral students majoring in coaching were asked to review the Q sorting in advance. Thorough preliminary Q sorting work, final checking of the Q statements, and preparations for Q sorting guidance work were carried out. Through this process, an explanation of the study’s purpose was prepared, and the rough Q statements were further adjusted.

After the preliminary preparation process, 40 statement cards selected as Q samples were presented to the P samples. Q sorting was performed for approximately 15 days from 1–15 April 2024. The P samples were asked to first sort the Q statement cards as positive, neutral, or negative. Then, they were asked to perform forced distribution tasks so that the 40 cards could be used to make a normal distribution. Based on the statement numbers recorded in the Q sample distribution chart, the list scored by ordering the items from the statement with which they most disagreed scored 1 Point and to the statement with which they most agreed scored 11 points. This information was then coded and entered into the Quanl program and analyzed and interpreted through principal component analysis. Each type was produced with an eigenvalue of 1 or higher, and the reasons for selecting the most agreed or disagreed with items were examined, focusing on the P sample with a high weight for each type.

## 3. Study Results

### 3.1. Result Analysis

A total of four types were derived from this study, as shown in Table 4. The eigenvalues by type were 8.2997 for type 1, 2.5996 for type 2, 2.4072 for type 3, and 1.9271 for type 4, and the cumulative value was 0.4914.

As shown in Table 5, the correlation coefficients that show the similarities between individual types were found to be 0.355 between type 1 and type 2, 0.375 between type 1 and type 3, 0.498 between type 1 and type 4, 0.316 between type 2 and type 3, 0.414 between type 2 and type 4, and 0.401 between type 3 and type 4.

Factor weights were categorized, and the results were obtained as shown in Table 6. In the case of type 1, P25 showed the highest factor weight at 1.7423. For type 2, P11 showed the highest factor weight at 1.3429. In the case of type 3, P21 showed the highest factor weight at 1.4041. Finally, for type 4, P8 showed the highest factor weight at 1.8810.

### 3.2. Perception Type Characteristics

#### 3.2.1. Type 1: A Perception Type That Considers Altruism to Lead to Respect for a Client’s Presence

Type 1 considers altruism as being able to lead to respect for the client’s presence and was named accordingly. As presented in Table 7, type 1 P samples showed the strongest agreement with Q11, ‘Altruistic feelings and actions make me happy’ (z = 1.54), followed by Q30, ‘Coach’s altruism makes him listen to the voice of clients’ (z = 1.47); Q15, ‘Altruism connects us to each other’ (z = 1.17); Q35, ‘Altruism makes us respect the existence of a person, not the condition of the person’ (z = 1.16); and Q3, ‘A coach’s altruism is an important element in forming relationships with clients’ (z = 1.11) Meanwhile, type 1 P samples showed the strongest disagreement with Q10, ‘I have an obsession to be altruistic when coaching’ (z = −2.42), followed by Q2, ‘If coaching is continued, altruism will be developed’ (z = −1.90); Q39, ‘If a coach does not have altruism, he or she should not coach’ (z = −1.89), and Q28, ‘The act of coaching is a manifestation of altruism’ (z = −1.67).

P25 (1.7423), who had the highest factor weight among type 1 P samples, stated the following: ‘When altruism and listening are combined, altruism becomes a good coaching tool. Altruism-based coaching enables excellent coaching. Coaches must pursue the interests of their clients through altruism, not their own interests’. P23 (1.5205), who had the second highest factor weight among type 1 P samples, stated, ‘Every person cherishes himself/herself the most and can easily understand himself/herself. Altruism makes me feel that I and the client are the same so that I can consider and understand the client with the same mind for me’.

#### 3.2.2. Type 2: A Perception Type That Considers Altruism to Be Meaningful When Practiced

Type 2 believes that altruism should be expressed and practiced through words and actions. As presented in Table 8, type 2 P samples showed the strongest agreement with Q5, ‘Altruism should not only be an emotional experience but should also be practiced’ (z = 1.68), followed by Q4, ‘Coaches’ altruism is manifested in their words and actions’ (z = 1.58); Q35, ‘Altruism makes us respect the existence of a person, not the condition of the person’ (z = 1.42); and Q30, ‘A coach’s altruism makes him listen to the voice of clients’ (z = 1.35). Meanwhile, type 2 P samples showed the strongest disagreement with Q22, ‘Altruism is the idea that another person is the same as me’ (z = −2.19), followed by Q34, ‘I think altruism is related to spirituality’ (z = −1.93), and Q10, ‘I have an obsession to be altruistic when coaching’ (z = −1.92).

P11 (1.3429), who had the highest factor weight among type 2 P samples, stated, ‘All good hearts are expressed purely when I am relaxed and stable. Helping and looking after others rather than feeling self-satisfaction when you are comfortable is altruism’. P19 (1.2323), who had the second highest factor weight among type 2 P samples, stated, ‘Altruism is basically taking one’s own losses and benefiting others. Like soldiers who sacrifice themselves to benefit others, altruism can be said to be true altruism only when it is accompanied by practice. To do so, we must naturally learn to be considerate and helpful to others from childhood’.

#### 3.2.3. Type 3: A Perception Type That Considers Altruism to Be a Mindset That Leads to the Development of Happiness

Type 3 feels that mutual happiness can occur through altruism. As shown in Table 9, type 3 P samples showed the strongest agreement with Q11, ‘Altruistic feelings and actions make me happy’ (z = 1.99), followed by Q19, ‘I usually feel altruism frequently’ (z = 1.38); Q33, ‘When I see a client who wants help, I feel altruistic’ (z = 1.37); and Q9, ‘Altruism is an inherent nature of humans’ (z = 1.35). Meanwhile, type 3 P samples showed the strongest disagreement with Q10, ‘I have an obsession to be altruistic when coaching’ (z = −2.42), followed by Q2, ‘If coaching is continued, altruism will be developed’ (z = −1.90); Q39, ‘If a coach does not have altruism, he or she should not coach’ (z = −2.29); and Q21, ‘Coaches must prioritize the interests of their clients over their own’ (z = −1.93).

P21 (1.4041), who had the highest factor weight among the type 3 P samples, stated, ‘The heart for others is innate. When I have a talk with clients, the higher my altruism, the more I experience rich and peaceful feelings’. P3 (1.1043), who had the second highest factor weight among the type 3 P samples, said, ‘Altruistic feelings and actions make me happy. When I have such feelings, if the people I am connected to feel good and happy, I am happier’.

#### 3.2.4. Type 4: A Perception Type That Considers Altruism to Be a Basic Attitude All Coaches Should Have

Type 4 recognizes altruism as a basic virtue that all coaches should have. As indicated in Table 10, type 4 P samples showed the strongest agreement with Q3, ‘A coach’s altruism is an important element in forming relationships with clients’ (z = 2.18), followed by Q20, ‘Altruism is a basic virtue of coaches (z = 1.60); Q4, ‘Coaches’ altruism is manifested in their words and actions’ (z = 1.45); and Q1, ‘I think coaches’ actions to support the development of their clients reflect altruism’ (z = 1.35). Meanwhile, type 3 P samples showed the strongest disagreement with Q10, ‘I have an obsession to be altruistic when coaching’ (z = −2.32), followed by Q22, ‘Altruism is the idea that another person is the same as me’ (z = −2.00), and Q24, ‘I feel altruism when I let go of my desires’ (z = −1.60).

P8 (1.8810), who had the highest factor weight among the type 4 P samples, stated, ‘When a coach wants to help a client grow, altruism is a mindset the coach basically must have, and such a mindset is essential for forming a bond of sympathy with the client. Altruism is an inherent part of basic human nature, and this nature as such is used in coaching as it is’. P30 (0.8553), who had the second highest factor weight among the type 4 P samples, said, ‘I think that coaches’ willingness to help clients regardless of the benefit, like altruism, has real value. Such sincerity is passed on to clients, too, thereby strengthening trust with them and making better coaching results expectable’.

### 3.3. Consensus Items

A total of 10 consensus items, which are items commonly agreed with by type, appeared as shown in Table 11. Analyzing other items and considering consensus items helps us understand the individual characteristics of each type.

## 4. Discussion and Conclusions

According to the results of this study, coaches’ perceptions of altruism were divided into four types: type 1, ‘a perception type that considers altruism to lead to respect for a client’s presence’; type 2, ‘a perception type that considers altruism to be meaningful when practiced’; type 3, ‘a perception type that considers altruism to be a mindset that leads to the development of happiness’; and type 4, ‘a perception type that considers altruism to be a basic attitude that coaches should have’.

Among the P samples, nine coaches corresponded to type 1. Some researchers consider altruism to be an important element in the formation of a therapeutic relationship [35,36,37], and type 1 coaches believe that altruism also makes coaches respect clients’ presence, thereby improving their understanding of their clients. In addition, by respecting the customer’s presence, coaches of this type feel they can improve their understanding of their clients.

Among the P samples, nine coaches corresponded to type 2. They believe that altruism should not only be an emotional experience but should also be practiced through words and actions. Smith et al. [1] stated that the tendency to participate in actions to help others (i.e., altruism) occurs after understanding the situation of others. Type 2 coaches believed that they should empathize with clients and express and practice the resulting altruism to help clients directly.

Among the P samples, six coaches corresponded to type 3. They frequently feel altruistic and believe that altruism makes them and their clients happy. Baston [9] defined altruism as a motivational state with the goal of promoting the happiness of others. Type 3 coaches believe that altruism makes them happy and willing to work for their clients’ development, ultimately making their clients happy as well.

Among the P samples, seven coaches corresponded to type 4. They believe that altruism is a basic human virtue and that coaches should treat clients altruistically. Swank et al. [38] stated that in counseling, the counselor must create a relationship in which they have altruistic consideration and interest without expecting any reward from the client. As such, type 4 coaches are also people who believe that altruism, which makes coaches have consideration for and interest in their clients, should be a basic virtue of coaches.

Based on Robinson and Curry’s [35] the Altruism Development Model (ADM), coaches’ perceptions of altruism were divided into biological, cognitive, social, and religious/spiritual dimensions, as shown in Table 12.

Type 1 was classified into the religious and spiritual dimension because type 1 coaches consider altruism to lead them toward respect for clients without any conditions. Altruism is a core element of most religious traditions [39], and individuals’ religious and spiritual beliefs influence the development of altruism [10]. As altruistic tendencies increase, the relevant persons are more concerned about the benefits of those who are helped rather than their efforts or costs involved in helping others [36]. Similar to type 1, the perception is that coaches will be of great help in achieving the common good by improving the mental well-being of society members so they can respect and help each other.

Type 2 was classified into the social dimension because type 2 coaches consider altruism to be meaningful when expressed and practiced. Actively expressing positive psychology increases perceptions of abundance [40]. In addition, members of successful groups are highly likely to instinctively conduct more considerate behaviors towards others [5]. A perception like that of type 2 coaches can help coaches and clients achieve common coaching goals.

Type 3 is classified into the cognitive dimension because type 3 coaches frequently feel altruistic and feel that altruism leads to feelings of happiness. For people who frequently feel positive emotions, happiness can increase significantly [41]. Schwartz et al. [42] stated that the act of helping others is related to good mental health, such as reduced distress and positive coping. Type 3 coaches are motivated to show care towards their clients through altruism, ultimately becoming happy themselves. Perspectives like those of type 3 coaches show that altruism is not only about serving others but that it can ultimately make you happy as well.

Type 4 was classified into the biological dimension because type 4 coaches consider altruism to be an innate attitude that is part of human nature. Hamilton [43] posited that humans have an innate altruistic gene, and type 4 coaches believe that coaches should naturally manifest humans’ innate altruism. Reflecting the same perception as that of type 4 coaches, altruism should be considered a basic qualification of coaches, and it should be reflected in coaches’ ethics and professional consciousness.

Previous studies have been conducted on the definition, concept, role, and evolutionary aspects of altruism, as well as models that can develop altruism; however, no studies have dealt with individuals’ subjective perceptions of altruism. Through this study, coaches’ subjective perceptions of and their opinions on altruism are examined. According to the results of this study, coaches perceive altruism as motivating them to respect their clients’ presence or that it is more meaningful when practiced. In addition, altruism was recognized as something that makes coaches feel happy and as a basic attitude that coaches should have. The coaches’ perceptions of altruism derived in this study were divided into biological, cognitive, social, and religious/spiritual categories based on the ADM developed by Robinson and Curry [35] and were examined more multidimensionally. Through this study, the researchers sought ways to make coaches happier, help the common good, and cultivate their professional consciousness and ethics through altruism. The study randomly recruited and selected participants to understand the varying opinions of coaches on altruism.

This study examined South Korean coaches’ perceptions of altruism using Q methodology; however, it has some limitations. First, it is unclear whether each type derived through Q methodology sufficiently reflects the diverse thoughts of various coaches. If more participants were interviewed and the age and occupation of the participants were more diversified, a greater variety of opinions on altruism could be obtained, thereby enriching the study. Further analysis is also needed of coaches’ internal experiences and ideas about altruism through in-depth qualitative studies. Accordingly, in-depth qualitative research on coaches’ altruism should be conducted using the phenomenological study method. Second, differences may exist between how coaches perceive altruism and how it is viewed by the public. Eisenberg et al. [7] stated that altruism is developed cognitively as individuals build up the ability to accept other people’s perspectives, and while conducting the study, it appeared that many coaches had concerns or thoughts about altruism even at normal times owing to their professional characteristics. Therefore, future studies should compare the degrees of altruism between coaches and the public to verify this quantitatively. Third, it is necessary to diversify the scope of the study sample by characteristics such as country, generation, and gender to further enrich perceptions of altruism and compare differences in perceptions according to participants.

## Figures and Tables

**Table 1 behavsci-14-00802-t001:** Study process.

Stage	Content
Stage 1: Organization of Q population	Related literature review, interviews, questionnaire surveys
Stage 2: Selection of Q samples	Q statement sorting and supplementation, N = 40
Stage 3: Composition of P samples	Coach group, N = 31
Stage 4: Q sorting and data analysis	Compulsory sorting by P samples and analyses by type

**Table 2 behavsci-14-00802-t002:** Interviewee information.

No	Gender	Coach Certification	Coaching Type	Job	Age
1	F	KAC	Business	Public servant	58
2	M	KPC	Business	Office worker	52
3	F	KAC	Business	Office worker	52
4	F	KPC	Career	Teacher	52
5	F	KPC	Life	Homemaker	47

**Table 3 behavsci-14-00802-t003:** Q statements.

Altruism Category	Q Statement
Cognitive	Q1. I think coaches’ actions to support the development of their clients reflect altruism.
Social	Q2. If coaching is continued, altruism will be developed.
Social	Q3. A coach’s altruism is an important element in forming relationships with clients.
Cognitive	Q4. Coaches’ altruism is manifested in their words and actions.
Social	Q5. Altruism should not only be an emotional experience but should also be practiced.
Social	Q6. Altruism can be developed through social learning.
Cognitive	Q7. Altruistic coaches have better empathic abilities.
Cognitive	Q8. Altruism improves coaches’ understanding of clients.
Biological	Q9. Altruism is inherent in human nature.
Cognitive	Q10. I have an obsession with being altruistic when coaching.
Cognitive	Q11. Altruistic feelings and actions make me happy.
Social	Q12. Altruism helps form empathy with clients.
Social	Q13. Altruism can be suppressed or promoted by environmental factors.
Religious/spiritual	Q14. If we give something to others, it will come back to us.
Religious/spiritual	Q15. Altruism connects us to each other.
Biological	Q16. If I am busy and tired, I can hardly be altruistic.
Cognitive	Q17. To be altruistic, a coach should have high self-esteem.
Cognitive	Q18. A coach must have the mindset to help clients without expecting anything in return.
Biological	Q19. I frequently feel altruistic.
Cognitive	Q20. Altruism is a basic virtue of coaches.
Social	Q21. Coaches must prioritize their clients’ interests over their own.
Cognitive	Q22. Altruism is the idea that another person is the same as me.
Religious/spiritual	Q23. When I exercise altruism, I feel that my mind is also healed.
Religious/spiritual	Q24. I feel altruistic when I let go of my desires.
Cognitive	Q25. When I behave altruistically, I feel like I become a good person.
Cognitive	Q26. If I think I am making a sacrifice, it is not altruism.
Biological	Q27. Altruism is also an act for my desires.
Cognitive	Q28. The act of coaching is a manifestation of altruism.
Cognitive	Q29. Excessive altruism can make the other person uncomfortable.
Cognitive	Q30. Altruism makes coaches listen to the voices of their clients.
Cognitive	Q31. Altruism enables coaches to devise ways to help clients.
Cognitive	Q32. A coach’s altruism helps the client’s self-awareness.
Cognitive	Q33. When I see a client who wants help, I feel altruistic.
Religious/spiritual	Q34. I think altruism is related to spirituality.
Religious/spiritual	Q35. Altruism makes us respect the existence of a person, not the condition of the person.
Social	Q36. People often act altruistically because they want to be recognized by others.
Cognitive	Q37. Altruism enables client-centered coaching.
Cognitive	Q38. When I listen to clients’ issues, I feel altruistic.
Social	Q39. If a coach is not altruistic, he or she should not coach.
Cognitive	Q40. Altruism makes coaches immerse themselves in the subject of coaching.

**Table 4 behavsci-14-00802-t004:** Eigenvalues and explanatory variances in the sorting of four types.

Content	I	II	III	IV
Chosen eigenvalues	8.2997	2.5996	2.4072	1.9271
Cumulative	0.2677	0.3516	0.4292	0.4914

**Table 5 behavsci-14-00802-t005:** Correlation coefficients between types.

Type	I	II	III	IV
I	1.000			
II	0.355	1.000		
III	0.375	0.316	1.000	
IV	0.498	0.414	0.401	1.000

**Table 6 behavsci-14-00802-t006:** P samples and factor weights by type.

No	Factor Loading	Gender	Coach Certification	Coaching Type	Job	Age
Type 1, N = 9						
P1	0.9600	M	KPC	Career	Graduate student	49
P4	0.9734	M	KPC	Career	Military officer	56
P10	0.9236	F	KPC	Career	Graduate student	38
P16	0.7061	F	Preparing for KAC	Life	Homemaker	52
P18	0.7916	F	KPC	Business	Professional coach	53
P23	1.5205	F	KAC	Business	Office worker	50
P25	1.7423	M	KAC	Business	Police officer	54
P27	0.8883	F	KAC	Career	Graduate student	28
P31	0.8711	F	KAC	Business	Office worker	50
Type 2, N = 9						
P11	1.3429	F	Preparing for KAC	Business	Office worker	28
P12	0.2330	M	KAC	Business	Office worker	29
P14	0.4772	F	KPC	Business	Professional coach	62
P17	0.9289	F	Preparing for KAC	Business	Office worker	43
P19	1.2323	F	KPC	Business	Professional coach	56
P20	0.6234	F	KPC	Business	Professional coach	54
P22	1.1752	F	KPC	Business	Office worker	48
P24	0.8493	M	KPC	Business	Office worker	54
P26	0.7944	F	Preparing for KAC	Career	Teacher	48
Type 3 (N = 6)						
P3	1.1043	F	KPC	Business	Office worker	46
P5	0.6217	F	KAC	Career	Teacher	52
P13	0.5867	F	KPC	Business	Office worker	48
P21	1.4041	F	KAC	Career	Military officer	40
P28	0.8115	F	Preparing for KAC	Career	Teacher	46
P29	0.6516	F	Preparing for KAC	Life	Pharmacist	49
Type 4 (N = 7)						
P2	0.5599	M	KPC	Business	Office worker	52
P6	0.7276	F	KPC	Business	Office worker	52
P7	0.6807	F	KPC	Business	Office worker	54
P8	1.8810	F	Preparing for KAC	Business	Public servant	56
P9	0.4416	F	KPC	Business	Professional Coach	52
P15	0.4868	F	KAC	Life	Homemaker	38
P30	0.8553	F	Preparing for KAC	Business	Office Worker	44

**Table 7 behavsci-14-00802-t007:** Statements and standard scores (at least ±1.00) of type 1.

No.	Statement	Standard Score
11	Altruistic feelings and actions make me happy.	1.54
30	A coach’s altruism makes him listen to the voice of clients.	1.47
15	Altruism connects us to each other.	1.17
35	Altruism makes us respect the existence of a person, not the condition of the person.	1.16
3	A coach’s altruism is an important element in forming relationships with clients.	1.11
23	When I exercise altruism, I feel that my mind is also healed.	1.05
8	Altruism improves coaches’ understanding of clients.	1.01
5	Altruism should not only be an emotional experience but should also be practiced.	1.00
36	People often act altruistically because they want to be recognized by others.	−1.33
26	If I think I am making a sacrifice, it is not altruism.	−1.62
28	The act of coaching is a manifestation of altruism.	−1.67
39	If a coach does not have altruism, he or she should not coach.	−1.89
2	If coaching is continued, altruism will be developed.	−1.90
10	I have an obsession to be altruistic when coaching.	−2.42

**Table 8 behavsci-14-00802-t008:** Statements and standard scores (at least ±1.00) of type 2.

No.	Statement	Standard Score
5	Altruism should not only be an emotional experience but should also be practiced.	1.68
4	Coaches’ altruism is manifested in their words and actions.	1.58
35	Altruism makes us respect the existence of a person, not the condition of the person.	1.42
30	Altruism makes coaches listen to the voices of their clients.	1.35
17	To be altruistic, a coach should have high self-esteem.	1.34
20	Altruism is a basic virtue of coaches.	1.17
26	If I think I am making a sacrifice, it is not altruism.	1.13
8	Altruism improves coaches’ understanding of clients.	1.02
29	Excessive altruism can make the other person uncomfortable.	−1.35
14	If we give something to others, it will come back to us.	−1.57
10	I have an obsession to be altruistic when coaching.	−1.92
34	I think altruism is related to spirituality.	−1.93
22	Altruism is the idea that another person is the same as me.	−2.19

**Table 9 behavsci-14-00802-t009:** Statements and standard scores (at least ±1.00) of type 3.

No.	Statement	Standard Score
11	Altruistic feelings and actions make me happy.	1.99
19	I usually feel altruism frequently.	1.38
33	When I see a client who wants help, I feel altruistic.	1.37
9	Altruism is an inherent nature of humans.	1.35
1	I think coaches’ actions to support the development of their clients reflect altruism.	1.20
13	Altruism can be suppressed or promoted by environmental factors.	1.08
24	I feel altruism when I let go of my desires.	−1.09
18	A coach must have the mind to help clients without expecting anything in return.	−1.09
34	I think altruism is related to spirituality.	−1.09
32	A coach’s altruism helps client’s self-awareness.	−1.09
29	Excessive altruism can make the other person uncomfortable.	−1.24
21	Coaches must prioritize the interests of their clients over their own.	−1.93
39	If a coach does not have altruism, he or she should not coach.	−2.29
10	I have an obsession to be altruistic when coaching.	−2.38

**Table 10 behavsci-14-00802-t010:** Statements and standard scores (at least ±1.00) of type 4.

No.	Statement	Standard Score
3	A coach’s altruism is an important element in forming relationships with clients.	2.18
20	Altruism is a basic virtue of coaches.	1.60
4	Coaches’ altruism is manifested in their words and actions.	1.45
1	I think coaches’ actions to support the development of their clients reflect altruism.	1.17
12	Altruism helps form empathy with clients.	1.05
34	I think altruism is related to spirituality.	−1.06
38	When I listen to clients’ issues, I feel altruistic.	−1.08
2	If coaching is continued, altruism will be developed.	−1.28
17	To be altruistic, a coach should have high self-esteem.	−1.55
24	I feel altruism when I let go of my desires.	−1.60
22	Altruism is the idea that another person is the same as me.	−2.00
10	I have an obsession to be altruistic when coaching.	−2.32

**Table 11 behavsci-14-00802-t011:** Consensus items of each type.

No.	Statement	Standard Score
35	Altruism makes us respect the existence of a person, not the condition of the person.	1.05
8	Altruism improves coaches’ understanding of clients.	0.79
12	Altruism helps form empathy with clients.	0.74
15	Altruism connects us to each other.	0.52
16	If I am busy and tired, I can hardly have altruism.	0.50
31	Altruism enables coaches to devise ways to help clients.	0.49
37	Altruism enables client-centered coaching.	0.40
40	Altruism makes coaches immerse themselves in the topic of coaching.	−0.08
36	People often act altruistically because they want to be recognized by others.	−1.00
10	I have an obsession to be altruistic when coaching.	−2.26

**Table 12 behavsci-14-00802-t012:** Individual types and division of dimensions.

	Biological	Cognitive	Social	Religious/Spiritual
Type 1: a perception type that considers altruism to lead to respect for the client’s presence				O
Type 2: a perception type that considers altruism to be meaningful when practiced			O	
Type 3: a perception type that considers altruism to be a mindset that leads to the development of happiness		O		
Type 4: a perception type that considers altruism to be a basic attitude that coaches should have	O			

## Data Availability

Data from the study are available upon request.

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
