# Peer review of "The Classification of Altruism Forms South Korean Coaches’ Perspective"

_behavsci, 2024, doi:10.3390/bs14090802_

Round 1

Reviewer 1 Report

Comments and Suggestions for Authors

Overall, I found this an engaging and well-written paper. I would like to haver seen more discussion of the implications for coach development and coach education.

An aspect of this which the authors might like to consider is that wisdom studies in recent years have included altruism as a key component of wisdom. Mentoring is closely associated with the use of and development of wisdom. And the work of the coach maturity research group finds that mature coaches integrate coaching and mentoring.

Reviewer 2 Report

Comments and Suggestions for Authors

The Article  "A Study on South Korean Coaches' Perceptions on Altruism" is well written and crisp. It nicely outlines different perceptions on individuals willingness to go the extra mile to benefit others through selflessness and mindfulness. The authors could have added one or two more factors instead of just 4 factors for a more comprehensive output. Also the references should be centered more on Altruism. Some relevant data analysis & charts on Altruism would make it a interesting read.

Reviewer 3 Report

Comments and Suggestions for Authors

In summary, lines 17-18: The results of this study will enhance coaches' attitudes towards clients and help them go further in developing their ethics and professional consciousness... the phrase seems extremely optimistic, I suggest the authors to be a bit more pragmatic, knowing the fact that this virtue (altruism) is related to education, conscience and maturity.

The hypotheses seem convoluted, I recommend the authors to write them much more simply, excluding the words frequency of repetitive words perceptions and altruism in the respective combinations.

The bibliographic sources used, the majority are not really current, they are more than 10 years old. In this sense, I recommend the authors to look for current sources and possibly modify/add in the Introduction, respectively in the Discussions, any new elements.
